# PARAMETER SPACE NOISE FOR EXPLORATION

**Matthias Plappert**[†‡]**, Rein Houthooft**[†]**, Prafulla Dhariwal**[†]**, Szymon Sidor**[†]**,**
**Richard Y. Chen**[†]**, Xi Chen**[††]**, Tamim Asfour**[‡]**, Pieter Abbeel**[††]**, and Marcin Andrychowicz**[†]

[†] OpenAI
[‡] Karlsruhe Institute of Technology (KIT)
[††] University of California, Berkeley
Correspondence to `matthias@openai.com`

## ABSTRACT

Deep reinforcement learning (RL) methods generally engage in exploratory behavior through noise injection in the action space. An alternative is to add noise directly to the agent's parameters, which can lead to more consistent exploration and a richer set of behaviors. Methods such as evolutionary strategies use parameter perturbations, but discard all temporal structure in the process and require significantly more samples. Combining parameter noise with traditional RL methods allows to combine the best of both worlds. We demonstrate that both off- and on-policy methods benefit from this approach through experimental comparison of DQN, DDPG, and TRPO on high-dimensional discrete action environments as well as continuous control tasks.

## 1 INTRODUCTION

Exploration remains a key challenge in contemporary deep reinforcement learning (RL). Its main purpose is to ensure that the agent's behavior does not converge prematurely to a local optimum. Enabling efficient and effective exploration is, however, not trivial since it is not directed by the reward function of the underlying Markov decision process (MDP). Although a plethora of methods have been proposed to tackle this challenge in high-dimensional and/or continuous-action MDPs, they often rely on complex additional structures such as counting tables (Tang et al., 2016), density modeling of the state space (Ostrovski et al., 2017), learned dynamics models (Houthooft et al., 2016; Achiam & Sastry, 2017; Stadie et al., 2015), or self-supervised curiosity (Pathak et al., 2017).

An orthogonal way of increasing the exploratory nature of these algorithms is through the addition of temporally-correlated noise, for example as done in bootstrapped DQN (Osband et al., 2016a). Along the same lines, it was shown that the addition of parameter noise leads to better exploration by obtaining a policy that exhibits a larger variety of behaviors (Sun et al., 2009b; Salimans et al., 2017). We discuss these related approaches in greater detail in Section 5. Their main limitation, however, is that they are either only proposed and evaluated for the on-policy setting with relatively small and shallow function approximators (Rückstieß et al., 2008) or disregard all temporal structure and gradient information (Salimans et al., 2017; Kober & Peters, 2008; Sehnke et al., 2010).

This paper investigates how parameter space noise can be effectively combined with off-the-shelf deep RL algorithms such as DQN (Mnih et al., 2015), DDPG (Lillicrap et al., 2015), and TRPO (Schulman et al., 2015b) to improve their exploratory behavior. Experiments show that this form of exploration is applicable to both high-dimensional discrete environments and continuous control tasks, using on- and off-policy methods. Our results indicate that parameter noise outperforms traditional action space noise-based baselines, especially in tasks where the reward signal is extremely sparse.

## 2 BACKGROUND

We consider the standard RL framework consisting of an agent interacting with an environment. To simplify the exposition we assume that the environment is fully observable. An environment is modeled as a Markov decision process (MDP) and is defined by a set of states $\mathcal{S}$, a set of actions $\mathcal{A}$, a distribution over initial states $p(s_0)$, a reward function $r : \mathcal{S} \times \mathcal{A} \mapsto \mathbb{R}$, transition probabilities

$p(s_{t+1}|s_t, a_t)$, a time horizon $T$, and a discount factor $\gamma \in [0, 1)$. We denote by $\pi_\theta$ a policy parametrized by $\theta$, which can be either deterministic, $\pi : \mathcal{S} \mapsto \mathcal{A}$, or stochastic, $\pi : \mathcal{S} \mapsto \mathcal{P}(\mathcal{A})$. The agent's goal is to maximize the expected discounted return $\eta(\pi_\theta) = \mathbb{E}_\tau[\sum_{t=0}^T \gamma^t r(s_t, a_t)]$, where $\tau = (s_0, a_0, \ldots, s_T)$ denotes a trajectory with $s_0 \sim p(s_0)$, $a_t \sim \pi_\theta(a_t|s_t)$, and $s_{t+1} \sim p(s_{t+1}|s_t, a_t)$. Experimental evaluation is based on the undiscounted return $\mathbb{E}_\tau[\sum_{t=0}^T r(s_t, a_t)]$.[1]

## 2.1 Off-policy Methods

Off-policy RL methods allow learning based on data captured by arbitrary policies. This paper considers two popular off-policy algorithms, namely Deep Q-Networks (DQN, Mnih et al. (2015)) and Deep Deterministic Policy Gradients (DDPG, Lillicrap et al. (2015)).

**Deep Q-Networks (DQN)** DQN uses a deep neural network as a function approximator to estimate the optimal $Q$-value function, which conforms to the Bellman optimality equation:

$$Q(s_t, a_t) = r(s_t, a_t) + \gamma \max_{a' \in \mathcal{A}} Q(s_{t+1}, a').$$

The policy is implicitly defined by $Q$ as $\pi(s_t) = \text{argmax}_{a' \in \mathcal{A}} Q(s_t, a')$. Typically, a stochastic $\epsilon$-greedy or Boltzmann policy (Sutton & Barto, 1998) is derived from the $Q$-value function to encourage exploration, which relies on sampling noise in the action space. The $Q$-network predicts a $Q$-value for each action and is updated using off-policy data from a replay buffer.

**Deep Deterministic Policy Gradients (DDPG)** DDPG is an actor-critic algorithm, applicable to continuous action spaces. Similar to DQN, the critic estimates the $Q$-value function using off-policy data and the recursive Bellman equation:

$$Q(s_t, a_t) = r(s_t, a_t) + \gamma Q\left(s_{t+1}, \pi_\theta(s_{t+1})\right),$$

where $\pi_\theta$ is the actor or policy. The actor is trained to maximize the critic's estimated $Q$-values by back-propagating through both networks. For exploration, DDPG uses a stochastic policy of the form $\widehat{\pi_\theta}(s_t) = \pi_\theta(s_t) + w$, where $w$ is either $w \sim \mathcal{N}(0, \sigma^2 I)$ (uncorrelated) or $w \sim \text{OU}(0, \sigma^2)$ (correlated).[2] Again, exploration is realized through action space noise.

## 2.2 On-policy Methods

In contrast to off-policy algorithms, on-policy methods require updating function approximators according to the currently followed policy. In particular, we will consider Trust Region Policy Optimization (TRPO, Schulman et al. (2015a)), an extension of traditional policy gradient methods (Williams, 1992b) using the natural gradient direction (Peters & Schaal, 2008; Kakade, 2001).

**Trust Region Policy Optimization (TRPO)** TRPO improves upon REINFORCE (Williams, 1992b) by computing an ascent direction that ensures a small change in the policy distribution. More specifically, TRPO solves the following constrained optimization problem:

$$\begin{aligned} \text{maximize}_\theta \quad & E_{s \sim \rho_{\theta'}, a \sim \pi_{\theta'}} \left[ \frac{\pi_\theta(a|s)}{\pi_\theta'(a|s)} A(s, a) \right] \\ \text{s.t.} \quad & E_{s \sim \rho_{\theta'}}[D_{\text{KL}}(\pi_{\theta'}(\cdot|s)\|\pi_\theta(\cdot|s))] \leq \delta_{\text{KL}} \end{aligned}$$

where $\rho_\theta = \rho_{\pi_\theta}$ is the discounted state-visitation frequencies induced by $\pi_\theta$, $A(s, a)$ denotes the advantage function estimated by the empirical return minus the baseline, and $\delta_{\text{KL}}$ is a step size parameter which controls how much the policy is allowed to change per iteration.

## 3 Parameter Space Noise for Exploration

This work considers policies that are realized as parameterized functions, which we denote as $\pi_\theta$, with $\theta$ being the parameter vector. We represent policies as neural networks but our technique can

---

[1] If $t = T$, we write $r(s_T, a_T)$ to denote the terminal reward, even though it has has no dependence on $a_T$, to simplify notation.

[2] OU($\cdot, \cdot$) denotes the Ornstein-Uhlenbeck process (Uhlenbeck & Ornstein, 1930).

be applied to arbitrary parametric models. To achieve structured exploration, we sample from a set of policies by applying additive Gaussian noise to the parameter vector of the current policy: $\widetilde{\theta} = \theta + \mathcal{N}(0, \sigma^2 I)$. Importantly, the perturbed policy is sampled at the beginning of each episode and kept fixed for the entire rollout. For convenience and readability, we denote this perturbed policy as $\widetilde{\pi} := \pi_{\widetilde{\theta}}$ and analogously define $\pi := \pi_{\theta}$.

**State-dependent exploration** As pointed out by Rückstieß et al. (2008), there is a crucial difference between action space noise and parameter space noise. Consider the continuous action space case. When using Gaussian action noise, actions are sampled according to some stochastic policy, generating $a_t = \pi(s_t) + \mathcal{N}(0, \sigma^2 I)$. Therefore, even for a *fixed* state $s$, we will almost certainly obtain a different action whenever that state is sampled again in the rollout, since action space noise is completely *independent* of the current state $s_t$ (notice that this is equally true for correlated action space noise). In contrast, if the parameters of the policy are perturbed at the beginning of each episode, we get $a_t = \widetilde{\pi}(s_t)$. In this case, the same action will be taken every time the same state $s_t$ is sampled in the rollout. This ensures consistency in actions, and directly introduces a dependence between the state and the exploratory action taken.

**Perturbing deep neural networks** It is not immediately obvious that deep neural networks, with potentially millions of parameters and complicated nonlinear interactions, can be perturbed in meaningful ways by applying spherical Gaussian noise. However, as recently shown by Salimans et al. (2017), a simple reparameterization of the network achieves exactly this. More concretely, we use layer normalization (Ba et al., 2016) between perturbed layers.[3] Due to this normalizing across activations within a layer, the same perturbation scale can be used across all layers, even though different layers may exhibit different sensitivities to noise.

**Adaptive noise scaling** Parameter space noise requires us to pick a suitable scale $\sigma$. This can be problematic since the scale will strongly depend on the specific network architecture, and is likely to vary over time as parameters become more sensitive to noise as learning progresses. Additionally, while it is easy to intuitively grasp the scale of action space noise, it is far harder to understand the scale in parameter space. We propose a simple solution that resolves all aforementioned limitations in an easy and straightforward way. This is achieved by adapting the scale of the parameter space noise over time and relating it to the variance in action space that it induces. More concretely, we can define a distance measure between perturbed and non-perturbed policy in action space and adaptively increase or decrease the parameter space noise depending on whether it is below or above a certain threshold:

$$\sigma_{k+1} = \begin{cases} \alpha \sigma_k & \text{if } d(\pi, \widetilde{\pi}) \leq \delta, \\ \frac{1}{\alpha} \sigma_k & \text{otherwise,} \end{cases} \tag{1}$$

where $\alpha \in \mathbb{R}_{>0}$ is a scaling factor and $\delta \in \mathbb{R}_{>0}$ a threshold value. The concrete realization of $d(\cdot, \cdot)$ depends on the algorithm at hand and we describe appropriate distance measures for DQN, DDPG, and TRPO in Appendix C.

**Parameter space noise for off-policy methods** In the off-policy case, parameter space noise can be applied straightforwardly since, by definition, data that was collected off-policy can be used. More concretely, we only perturb the policy for exploration and train the non-perturbed network on this data by replaying it.

**Parameter space noise for on-policy methods** Parameter noise can be incorporated in an on-policy setting, using an adapted policy gradient, as set forth by Rückstieß et al. (2008). Policy gradient methods optimize $\mathbb{E}_{\tau \sim (\pi, p)}[R(\tau)]$. Given a stochastic policy $\pi_\theta(a|s)$ with $\theta \sim \mathcal{N}(\phi, \Sigma)$, the expected return can be expanded using likelihood ratios and the re-parametrization trick (Kingma & Welling, 2013) as

$$\nabla_{\phi, \Sigma} \mathbb{E}_\tau[R(\tau)] \approx \frac{1}{N} \sum_{\epsilon^i, \tau^i} \left[ \sum_{t=0}^{T-1} \nabla_{\phi, \Sigma} \log \pi(a_t|s_t; \phi + \epsilon^i \Sigma^{\frac{1}{2}}) R_t(\tau^i) \right] \tag{2}$$

---

[3]This is in contrast to Salimans et al. (2017), who use virtual batch normalization, which we found to perform less consistently

for $N$ samples $\epsilon^i \sim \mathcal{N}(0, I)$ and $\tau^i \sim (\pi_{\phi + \epsilon^i \Sigma^{\frac{1}{2}}}, p)$ (see Appendix B for a full derivation). Rather than updating $\Sigma$ according to the previously derived policy gradient, we fix its value to $\sigma^2 I$ and scale it adaptively as described in Appendix C.

## 4 EXPERIMENTS

This section answers the following questions:

    (i) Do existing state-of-the-art RL algorithms benefit from incorporating parameter space noise?

    (ii) Does parameter space noise aid in exploring sparse reward environments more effectively?

    (iii) How does parameter space noise exploration compare against evolution strategies for deep policies (Salimans et al., 2017) with respect to sample efficiency?

Reference implementations of DQN and DDPG with adaptive parameter space noise are available online.[4]

### 4.1 COMPARING PARAMETER SPACE NOISE TO ACTION SPACE NOISE

The added value of parameter space noise over action space noise is measured on both high-dimensional discrete-action environments and continuous control tasks. For the discrete environments, comparisons are made using DQN, while DDPG and TRPO are used on the continuous control tasks.

**Discrete-action environments** For discrete-action environments, we use the Arcade Learning Environment (ALE, Bellemare et al. (2013)) benchmark along with a standard DQN implementation. We compare a baseline DQN agent with $\epsilon$-greedy action noise against a version of DQN with parameter noise. We linearly anneal $\epsilon$ from $1.0$ to $0.1$ over the first 1 million timesteps. For parameter noise, we adapt the scale using a simple heuristic that increases the scale if the KL divergence between perturbed and non-perturbed policy is less than the KL divergence between greedy and $\epsilon$-greedy policy and decreases it otherwise (see Section C.1 for details). By using this approach, we achieve a fair comparison between action space noise and parameter space noise since the magnitude of the noise is similar and also avoid the introduction of an additional hyperparameter.

For parameter perturbation, we found it useful to reparametrize the network in terms of an explicit policy that represents the greedy policy $\pi$ implied by the $Q$-values, rather than perturbing the $Q$-function directly. To represent the policy $\pi(a|s)$, we add a single fully connected layer after the convolutional part of the network, followed by a softmax output layer. Thus, $\pi$ predicts a discrete probability distribution over actions, given a state. We find that perturbing $\pi$ instead of $Q$ results in more meaningful changes since we now define an explicit behavioral policy. In this setting, the $Q$-network is trained according to standard DQN practices. The policy $\pi$ is trained by maximizing the probability of outputting the greedy action accordingly to the current $Q$-network. Essentially, the policy is trained to exhibit the same behavior as running greedy DQN. To rule out this double-headed version of DQN alone exhibits significantly different behavior, we always compare our parameter space noise approach against two baselines, regular DQN and two-headed DQN, both with $\epsilon$-greedy exploration.

We furthermore randomly sample actions for the first 50 thousand timesteps in all cases to fill the replay buffer before starting training. Moreover, we found that parameter space noise performs better if it is combined with a bit of action space noise (we use a $\epsilon$-greedy behavioral policy with $\epsilon = 0.01$ for the parameter space noise experiments). Full experimental details are described in Section A.1.

We chose 21 games of varying complexity, according to the taxonomy presented by (Bellemare et al., 2016). The learning curves are shown in Figure 1 for a selection of games (see Appendix D for full results). Each agent is trained for $40\,\mathrm{M}$ frames. The overall performance is estimated by running each configuration with three different random seeds, and we plot the median return (line) as well as the interquartile range (shaded area). Note that performance is evaluated on the exploratory policy since we are interested in its behavior especially.

---

    [4]`https://github.com/openai/baselines`

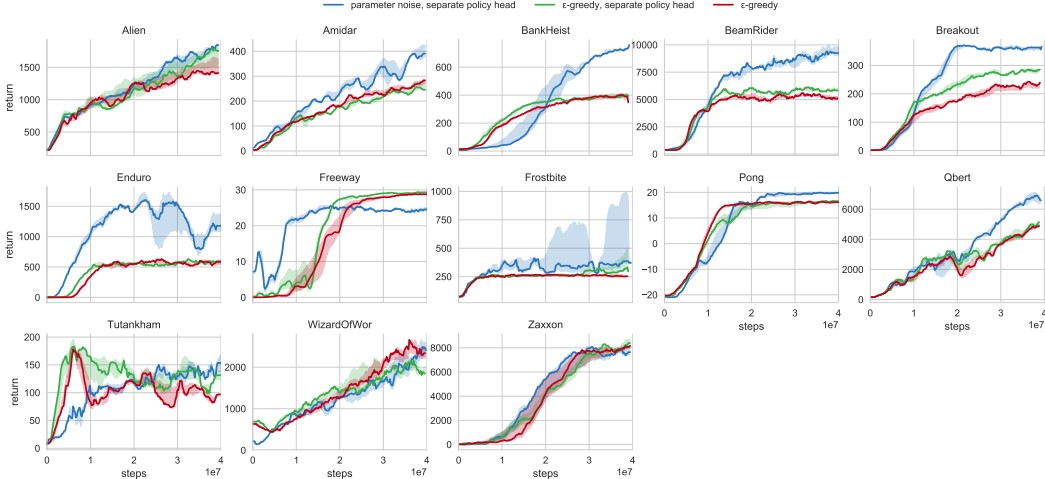

Figure 1: Median DQN returns for several ALE environment plotted over training steps.

Overall, our results show that parameter space noise often outperforms action space noise, especially on games that require consistency (e.g. Enduro, Freeway) and performs comparably on the remaining ones. Additionally, learning progress usually starts much sooner when using parameter space noise. Finally, we also compare against a double-headed version of DQN with $\epsilon$-greedy exploration to ensure that this change in architecture is not responsible for improved exploration, which our results confirm. Full results are available in Appendix D.

That being said, parameter space noise is unable to sufficiently explore in extremely challenging games like Montezuma's Revenge. More sophisticated exploration methods like Bellemare et al. (2016) are likely necessary to successfully learn these games. However, such methods often rely on some form of "inner" exploration method, which is usually traditional action space noise. It would be interesting to evaluate the effect of parameter space noise when combined with exploration methods.

On a final note, proposed improvements to DQN like double DQN (Hasselt, 2010), prioritized experience replay (Schaul et al., 2015), and dueling networks (Wang et al., 2015) are orthogonal to our improvements and would therefore likely improve results further. We leave the experimental validation of this theory to future work.

**Continuous control environments**  We now compare parameter noise with action noise on the continuous control environments implemented in OpenAI Gym (Brockman et al., 2016). We use DDPG (Lillicrap et al., 2015) as the RL algorithm for all environments with similar hyperparameters as outlined in the original paper except for the fact that layer normalization (Ba et al., 2016) is applied after each layer before the nonlinearity, which we found to be useful in either case and especially important for parameter space noise.

We compare the performance of the following configurations: (a) no noise at all, (b) uncorrelated additive Gaussian action space noise ($\sigma = 0.2$), (c) correlated additive Gaussian action space noise (Ornstein–Uhlenbeck process (Uhlenbeck & Ornstein, 1930) with $\sigma = 0.2$), and (d) adaptive parameter space noise. In the case of parameter space noise, we adapt the scale so that the resulting change in action space is comparable to our baselines with uncorrelated Gaussian action space noise (see Section C.2 for full details).

We evaluate the performance on several continuous control tasks. Figure 2 depicts the results for three exemplary environments. Each agent is trained for $1\,\text{M}$ timesteps, where $1$ epoch consists of $10$ thousand timesteps. In order to make results comparable between configurations, we evaluate the performance of the agent every $10$ thousand steps by using no noise for $20$ episodes.

On *HalfCheetah*, parameter space noise achieves significantly higher returns than all other configurations. We find that, in this environment, all other exploration schemes quickly converge to a local optimum (in which the agent learns to flip on its back and then "wiggles" its way forward). Parameter

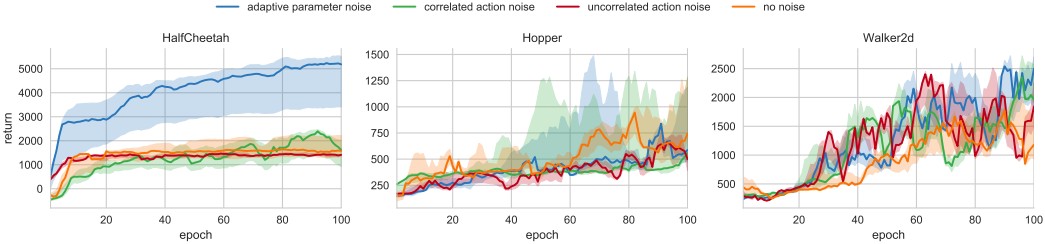

Figure 2: Median DDPG returns for continuous control environments plotted over epochs.

space noise behaves similarly initially but still explores other options and quickly learns to break out of this sub-optimal behavior. Also notice that parameter space noise vastly outperforms correlated action space noise on this environment, clearly indicating that there is a significant difference between the two. On the remaining two environments, parameter space noise performs on par with other exploration strategies. Notice, however, that even if no noise is present, DDPG is capable of learning good policies. We find that this is representative for the remaining environments (see Appendix E for full results), which indicates that these environments do not require a lot of exploration to begin with due to their well-shaped reward function.

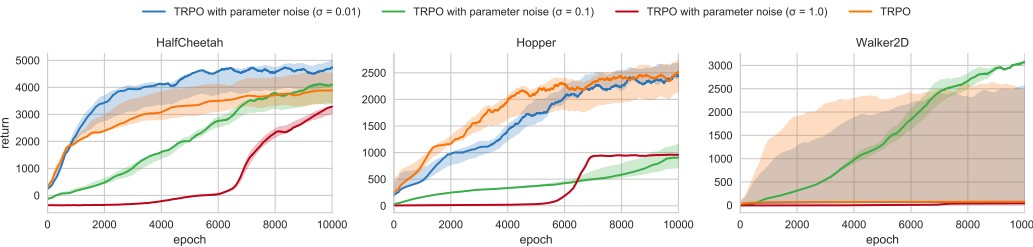

Figure 3: Median TRPO returns for continuous control environments plotted over epochs.

The results for TRPO are depicted in Figure 3. Interestingly, in the *Walker2D* environment, we see that adding parameter noise decreases the performance variance between seeds. This indicates that parameter noise aids in escaping local optima.

## 4.2 DOES PARAMETER SPACE NOISE EXPLORE EFFICIENTLY?

The environments in the previous section required relatively little exploration. In this section, we evaluate whether parameter noise enables existing RL algorithms to learn on environments with very sparse rewards, where uncorrelated action noise generally fails (Osband et al., 2016a; Achiam & Sastry, 2017).

**A scalable toy example** We first evaluate parameter noise on a well-known toy problem, following the setup described by Osband et al. (2016a) as closely as possible. The environment consists of a chain of $N$ states and the agent always starts in state $s_2$, from where it can either move left or right. In state $s_1$, the agent receives a small reward of $r = 0.001$ and a larger reward $r = 1$ in state $s_N$. Obviously, it is much easier to discover the small reward in $s_1$ than the large reward in $s_N$, with increasing difficulty as $N$ grows. The environment is described in greater detail in Section A.3.

We compare adaptive parameter space noise DQN, bootstrapped DQN, and $\epsilon$-greedy DQN. The chain length $N$ is varied and for each $N$ three different seeds are trained and evaluated. After each episode, we evaluate the performance of the current policy by performing a rollout with all noise disabled (in the case of bootstrapped DQN, we perform majority voting over all heads). The problem is considered solved if one hundred subsequent rollouts achieve the optimal return. We plot the median number of episodes before the problem is considered solved (we abort if the problem is still unsolved after 2 thousand episodes). Full experimental details are available in Section A.3.

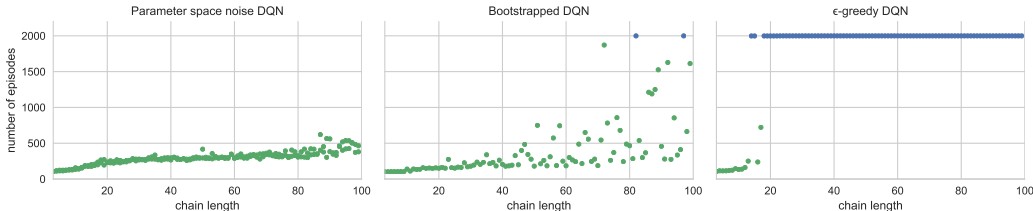

Figure 4: Median number of episodes before considered solved for DQN with different exploration strategies. Green indicates that the problem was solved whereas blue indicates that no solution was found within 2 K episodes. Note that less number of episodes before solved is better.

Figure 4 shows that parameter space noise clearly outperforms action space noise (which completely fails for moderately large $N$) and even outperforms the more computational expensive bootstrapped DQN. However, it is important to note that this environment is extremely simple in the sense that the optimal strategy is to always go right. In a case where the agent needs to select a different optimal action depending on the current state, parameter space noise would likely work less well since weight randomization of the policy is less likely to yield this behavior. Our results thus only highlight the difference in exploration behavior compared to action space noise in this specific case. In the general case, parameter space noise does not guarantee optimal exploration.

**Continuous control with sparse rewards**   We now make the continuous control environments more challenging for exploration. Instead of providing a reward at every timestep, we use environments that only yield a non-zero reward after significant progress towards a goal. More concretely, we consider the following environments from rllab[5] (Duan et al., 2016), modified according to Houthooft et al. (2016): (a) *SparseCartpoleSwingup*, which only yields a reward if the paddle is raised above a given threshold, (b) *SparseDoublePendulum*, which only yields a reward if the agent reaches the upright position, and (c) *SparseHalfCheetah*, which only yields a reward if the agent crosses a target distance, (d) *SparseMountainCar*, which only yields a reward if the agent drives up the hill, (e) *SwimmerGather*, yields a positive or negative reward upon reaching targets. For all tasks, we use a time horizon of $T = 500$ steps before resetting.

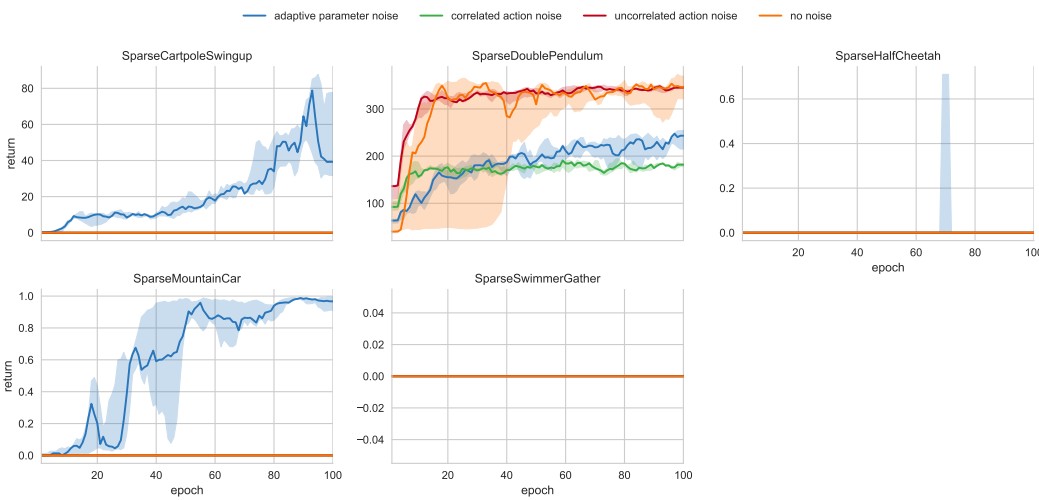

Figure 5: Median DDPG returns for environments with sparse rewards plotted over epochs.

We consider both DDPG and TRPO to solve these environments (the exact experimental setup is described in Section A.2). Figure 5 shows the performance of DDPG, while the results for TRPO have been moved to Appendix F. The overall performance is estimated by running each configuration with

---

[5]https://github.com/openai/rllab

five different random seeds, after which we plot the median return (line) as well as the interquartile range (shaded area).

For DDPG, *SparseDoublePendulum* seems to be easy to solve in general, with even no noise finding a successful policy relatively quickly. The results for *SparseCartpoleSwingup* and *SparseMountainCar* are more interesting: Here, only parameter space noise is capable of learning successful policies since all other forms of noise, including correlated action space noise, never find states with non-zero rewards. For *SparseHalfCheetah*, DDPG at least finds the non-zero reward but never learns a successful policy from that signal. On the challenging *SwimmerGather* task, all configurations of DDPG fail.

Our results clearly show that parameter space noise can be used to improve the exploration behavior of these off-the-shelf algorithms. However, it is important to note that improvements in exploration are not guaranteed for the general case. It is therefore necessary to evaluate the potential benefit of parameter space noise on a case-by-case basis.

### 4.3    Is RL with Parameter Space Noise more Sample-efficient than ES?

Evolution strategies (ES) are closely related to our approach since both explore by introducing noise in the parameter space, which can lead to improved exploration behavior (Salimans et al., 2017).[6] However, ES disregards temporal information and uses black-box optimization to train the neural network. By combining parameter space noise with traditional RL algorithms, we can include temporal information as well rely on gradients computed by back-propagation for optimization while still benefiting from improved exploratory behavior. We now compare ES and traditional RL with parameter space noise directly.

We compare performance on the 21 ALE games that were used in Section 4.1. The performance is estimated by running 10 episodes for each seed using the final policy with exploration disabled and computing the median returns. For ES, we use the results obtained by Salimans et al. (2017), which were obtained after training on $1\,000\,\mathrm{M}$ frames. For DQN, we use the same parameter space noise for exploration that was previously described and train on $40\,\mathrm{M}$ frames. Even though DQN with parameter space noise has been exposed to 25 times less data, it outperforms ES on 15 out of 21 Atari games (full results are available in Appendix D). Combined with the previously described results, this demonstrates that parameter space noise combines the desirable exploration properties of ES with the sample efficiency of traditional RL.

## 5    Related Work

The problem of exploration in reinforcement has been studied extensively. A range of algorithms (Kearns & Singh, 2002; Brafman & Tennenholtz, 2002; Auer et al., 2008) have been proposed that guarantee near-optimal solutions after a number of steps that are polynomial in the number of states, number of actions, and the horizon time. However, in many real-world reinforcements learning problems both the state and action space are continuous and high dimensional so that, even with discretization, these algorithms become impractical. In the context of deep reinforcement learning, a large variety of techniques have been proposed to improve exploration (Stadie et al., 2015; Houthooft et al., 2016; Tang et al., 2016; Osband et al., 2016a; Ostrovski et al., 2017; Sukhbaatar et al., 2017; Osband et al., 2016b). However, all are non-trivial to implement and are often computational expensive.

The idea of perturbing the parameters of a policy has been proposed by Rückstieß et al. (2008) for policy gradient methods. The authors show that this form of perturbation generally outperforms random exploration and evaluate their exploration strategy with the REINFORCE (Williams, 1992a) and Natural Actor-Critic (Peters & Schaal, 2008) algorithms. However, their policies are relatively low-dimensional compared to modern deep architectures, they use environments with low-dimensional state spaces, and their contribution is strictly limited to the policy gradient case. In contrast, our

---

[6]To clarify, when we refer to ES in this context, we refer to the recent work by Salimans et al. (2017), which demonstrates that deep policy networks that learn from pixels can be trained using ES. We understand that there is a vast body of other work in this field (compare section 5).

method is applied and evaluated for both on and off-policy setting, we use high-dimensional policies, and environments with large state spaces.

Our work is also closely related to evolution strategies (ES, Rechenberg & Eigen (1973); Schwefel (1977)), and especially neural evolution strategies (NES, Sun et al. (2009a;b); Glasmachers et al. (2010a;b); Schaul et al. (2011); Wierstra et al. (2014)). In the context of policy optimization, our work is closely related to Kober & Peters (2008) and Sehnke et al. (2010). More recently, Salimans et al. (2017) showed that ES can work for high-dimensional environments like Atari and OpenAI Gym continuous control problems. However, ES generally disregards any temporal structure that may be present in trajectories and typically suffers from sample inefficiency.

Bootstrapped DQN (Osband et al., 2016a) has been proposed to aid with more directed and consistent exploration by using a network with multiple heads, where one specific head is selected at the beginning of each episode. In contrast, our approach perturbs the parameters of the network directly, thus achieving similar yet simpler (and as shown in Section 4.2, sometimes superior) exploration behavior. Concurrently to our work, Fortunato et al. (2017) have proposed a similar approach that utilizes parameter perturbations for more efficient exploration.

## 6  CONCLUSION

In this work, we propose parameter space noise as a conceptually simple yet effective replacement for traditional action space noise like $\epsilon$-greedy and additive Gaussian noise. This work shows that parameter perturbations can successfully be combined with contemporary on- and off-policy deep RL algorithms such as DQN, DDPG, and TRPO and often results in improved performance compared to action noise. Experimental results further demonstrate that using parameter noise allows solving environments with very sparse rewards, in which action noise is unlikely to succeed. Our results indicate that parameter space noise is a viable and interesting alternative to action space noise, which is still the *de facto* standard in most reinforcement learning applications.

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

## A  EXPERIMENTAL SETUP

### A.1  ARCADE LEARNING ENVIRONMENT (ALE)

For ALE (Bellemare et al., 2013), the network architecture as described in Mnih et al. (2015) is used. This consists of 3 convolutional layers (32 filters of size $8 \times 8$ and stride 4, 64 filters of size $4 \times 4$ and stride 2, 64 filters of size $3 \times 3$ and stride 1) followed by 1 hidden layer with 512 units followed by a linear output layer with one unit for each action. ReLUs are used in each layer, while layer normalization (Ba et al., 2016) is used in the fully connected part of the network. For parameter space noise, we also include a second head after the convolutional stack of layers. This head determines a policy network with the same architecture as the $Q$-value network, except for a softmax output layer. The target networks are updated every $10 \, \mathrm{K}$ timesteps. The $Q$-value network is trained using the Adam optimizer (Kingma & Ba, 2015) with a learning rate of $10^{-4}$ and a batch size of 32. The replay buffer can hold $1 \, \mathrm{M}$ state transitions. For the $\epsilon$-greedy baseline, we linearly anneal $\epsilon$ from 1 to $0.1$ over the first $1 \, \mathrm{M}$ timesteps. For parameter space noise, we adaptively scale the noise to have a similar effect in action space (see Section C.1 for details), effectively ensuring that the maximum KL divergence between perturbed and non-perturbed $\pi$ is softly enforced. The policy is perturbed at the beginning of each episode and the standard deviation is adapted as described in Appendix C every 50 timesteps. Notice that we only perturb the policy head after the convolutional part of the network (i.e. the fully connected part, which is also why we only include layer normalization in this part of the network). To avoid getting stuck (which can potentially happen for a perturbed policy), we also use $\epsilon$-greedy action selection with $\epsilon = 0.01$. In all cases, we perform $50 \, \mathrm{K}$ random actions to collect initial data for the replay buffer before training starts. We set $\gamma = 0.99$, clip rewards to be in $[-1, 1]$, and clip gradients for the output layer of $Q$ to be within $[-1, 1]$. For observations, each frame is down-sampled to $84 \times 84$ pixels, after which it is converted to grayscale. The actual observation to the network consists of a concatenation of 4 subsequent frames. Additionally, we use up to 30 noop actions at the beginning of the episode. This setup is identical to what is described by Mnih et al. (2015).

### A.2  CONTINUOUS CONTROL

For DDPG, we use a similar network architecture as described by Lillicrap et al. (2015): both the actor and critic use 2 hidden layers with 64 ReLU units each. For the critic, actions are not included until the second hidden layer. Layer normalization (Ba et al., 2016) is applied to all layers. The target networks are soft-updated with $\tau = 0.001$. The critic is trained with a learning rate of $10^{-3}$ while the actor uses a learning rate of $10^{-4}$. Both actor and critic are updated using the Adam optimizer (Kingma & Ba, 2015) with batch sizes of 128. The critic is regularized using an $L2$ penalty with $10^{-2}$. The replay buffer holds $100 \, \mathrm{K}$ state transitions and $\gamma = 0.99$ is used. Each observation dimension is normalized by an online estimate of the mean and variance. For parameter space noise with DDPG, we adaptively scale the noise to be comparable to the respective action space noise (see Section C.2). For dense environments, we use action space noise with $\sigma = 0.2$ (and a comparable adaptive noise scale). Sparse environments use an action space noise with $\sigma = 0.6$ (and a comparable adaptive noise scale).

TRPO uses a step size of $\delta_{\mathrm{KL}} = 0.01$, a policy network of 2 hidden layers with 32 `tanh` units for the nonlocomotion tasks, and 2 hidden layers of 64 `tanh` units for the locomotion tasks. The Hessian calculation is subsampled with a factor of $0.1$, $\gamma = 0.99$, and the batch size per epoch is set to $5 \, \mathrm{K}$ timesteps. The baseline is a learned linear transformation of the observations.

The following environments from OpenAI Gym[7] (Brockman et al., 2016) are used:

- *HalfCheetah* ($\mathcal{S} \subset \mathbb{R}^{17}$, $\mathcal{A} \subset \mathbb{R}^6$),
- *Hopper* ($\mathcal{S} \subset \mathbb{R}^{11}$, $\mathcal{A} \subset \mathbb{R}^3$),
- *InvertedDoublePendulum* ($\mathcal{S} \subset \mathbb{R}^{11}$, $\mathcal{A} \subset \mathbb{R}$),
- *InvertedPendulum* ($\mathcal{S} \subset \mathbb{R}^4$, $\mathcal{A} \subset \mathbb{R}$),
- *Reacher* ($\mathcal{S} \subset \mathbb{R}^{11}$, $\mathcal{A} \subset \mathbb{R}^2$),
- *Swimmer* ($\mathcal{S} \subset \mathbb{R}^8$, $\mathcal{A} \subset \mathbb{R}^2$), and
- *Walker2D* ($\mathcal{S} \subset \mathbb{R}^{17}$, $\mathcal{A} \subset \mathbb{R}^6$).

For the sparse tasks, we use the following environments from rllab[8] (Duan et al., 2016), modified as described by Houthooft et al. (2016):

- *SparseCartpoleSwingup* ($\mathcal{S} \subset \mathbb{R}^4$, $\mathcal{A} \subset \mathbb{R}$), which only yields a reward if the paddle is raised above a given threshold,
- *SparseHalfCheetah* ($\mathcal{S} \subset \mathbb{R}^{17}$, $\mathcal{A} \subset \mathbb{R}^6$), which only yields a reward if the agent crosses a distance threshold,
- *SparseMountainCar* ($\mathcal{S} \subset \mathbb{R}^2$, $\mathcal{A} \subset \mathbb{R}$), which only yields a reward if the agent drives up the hill,
- *SparseDoublePendulum* ($\mathcal{S} \subset \mathbb{R}^6$, $\mathcal{A} \subset \mathbb{R}$), which only yields a reward if the agent reaches the upright position, and
- *SwimmerGather* ($\mathcal{S} \subset \mathbb{R}^{33}$, $\mathcal{A} \subset \mathbb{R}^2$), which yields a positive or negative reward upon reaching targets.

### A.3 CHAIN ENVIRONMENT

We follow the state encoding proposed by Osband et al. (2016a) and use $\phi(s_t) = (\mathbb{1}\{x \leq s_t\})$ as the observation, where $\mathbb{1}$ denotes the indicator function. DQN is used with a very simple network to approximate the $Q$-value function that consists of 2 hidden layers with 16 ReLU units. Layer normalization (Ba et al., 2016) is used for all hidden layers before applying the nonlinearity. Each agent is then trained for up to 2 K episodes. The chain length $N$ is varied and for each $N$ three different seeds are trained and evaluated. After each episode, the performance of the current policy is evaluated by sampling a trajectory with noise disabled (in the case of bootstrapped DQN, majority voting over all heads is performed). The problem is considered solved if one hundred subsequent trajectories achieve the optimal episode return. Figure 6 depicts the environment.

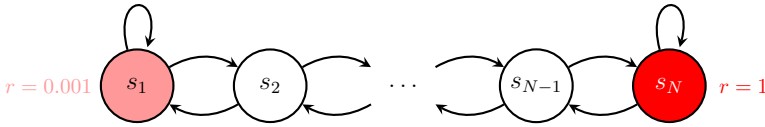

Figure 6: Simple and scalable environment to test for exploratory behavior (Osband et al., 2016a).

We compare adaptive parameter space noise DQN, bootstrapped DQN (Osband et al., 2016a) (with $K = 20$ heads and Bernoulli masking with $p = 0.5$), and $\epsilon$-greedy DQN (with $\epsilon$ linearly annealed from 1.0 to 0.1 over the first one hundred episodes). For adaptive parameter space noise, we only use a single head and perturb $Q$ directly, which works well in this setting. Parameter space noise is adaptively scaled so that $\delta \approx 0.05$. In all cases, $\gamma = 0.999$, the replay buffer holds 100 K state transitions, learning starts after 5 initial episodes, the target network is updated every 100 timesteps, and the network is trained using the Adam optimizer (Kingma & Ba, 2015) with a learning rate of $10^{-3}$ and a batch size of 32.

---

[7] https://github.com/openai/gym
[8] https://github.com/openai/rllab

## B    PARAMETER SPACE NOISE FOR ON-POLICY METHODS

Policy gradient methods optimize $\mathbb{E}_{\tau \sim (\pi, p)}[R(\tau)]$. Given a stochastic policy $\pi_\theta(a|s)$ with $\theta \sim \mathcal{N}(\phi, \Sigma)$, the expected return can be expanded using likelihood ratios and the reparametrization trick (Kingma & Welling, 2013) as

$$\nabla_{\phi, \Sigma} \mathbb{E}_\tau[R(\tau)] = \nabla_{\phi, \Sigma} \mathbb{E}_{\theta \sim \mathcal{N}(\phi, \Sigma)} \left[ \sum_\tau p(\tau|\theta) R(\tau) \right] \tag{3}$$

$$= \mathbb{E}_{\epsilon \sim \mathcal{N}(0, I)} \nabla_{\phi, \Sigma} \left[ \sum_\tau p(\tau|\phi + \epsilon \Sigma^{\frac{1}{2}}) R(\tau) \right] \tag{4}$$

$$= \mathbb{E}_{\epsilon \sim \mathcal{N}(0, I), \tau} \left[ \sum_{t=0}^{T-1} \nabla_{\phi, \Sigma} \log \pi(a_t|s_t; \phi + \epsilon \Sigma^{\frac{1}{2}}) R_t(\tau) \right] \tag{5}$$

$$\approx \frac{1}{N} \sum_{\epsilon^i, \tau^i} \left[ \sum_{t=0}^{T-1} \nabla_{\phi, \Sigma} \log \pi(a_t|s_t; \phi + \epsilon^i \Sigma^{\frac{1}{2}}) R_t(\tau^i) \right] \tag{6}$$

for $N$ samples $\epsilon^i \sim \mathcal{N}(0, I)$ and $\tau^i \sim (\pi_{\phi + \epsilon^i \Sigma^{\frac{1}{2}}}, p)$, with $R_t(\tau^i) = \sum_{t'=t}^{T} \gamma^{t'-t} r_{t'}^i$. This also allows us to subtract a variance-reducing baseline $b_t^i$, leading to

$$\nabla_{\phi, \Sigma} \mathbb{E}_\tau[R(\tau)] \approx \frac{1}{N} \sum_{\epsilon^i, \tau^i} \left[ \sum_{t=0}^{T-1} \nabla_{\phi, \Sigma} \log \pi(a_t|s_t; \phi + \epsilon^i \Sigma^{\frac{1}{2}}) (R_t(\tau^i) - b_t^i) \right]. \tag{7}$$

In our case, we set $\Sigma := \sigma^2 I$ and use our proposed adaption method to re-scale as appropriate.

## C    ADAPTIVE SCALING

Parameter space noise requires us to pick a suitable scale $\sigma$. This can be problematic since the scale will highly depend on the specific network architecture, and is likely to vary over time as parameters become more sensitive as learning progresses. Additionally, while it is easy to intuitively grasp the scale of action space noise, it is far harder to understand the scale in parameter space.

We propose a simple solution that resolves all aforementioned limitations in an easy and straight-forward way. This is achieved by adapting the scale of the parameter space noise over time, thus using a time-varying scale $\sigma_k$. Furthermore, $\sigma_k$ is related to the action space variance that it induces, and updated accordingly. Concretely, we use the following simple heuristic to update $\sigma_k$ every $K$ timesteps:

$$\sigma_{k+1} = \begin{cases} \alpha \sigma_k, & \text{if } d(\pi, \widetilde{\pi}) < \delta \\ \frac{1}{\alpha} \sigma_k, & \text{otherwise,} \end{cases} \tag{8}$$

where $d(\cdot, \cdot)$ denotes some distance between the non-perturbed and perturbed policy (thus measuring in action space), $\alpha \in \mathbb{R}_{>0}$ is used to rescale $\sigma_k$, and $\delta \in \mathbb{R}_{>0}$ denotes some threshold value. This idea is based on the Levenberg-Marquardt heuristic (Ranganathan, 2004). The concrete distance measure and appropriate choice of $\delta$ depends on the policy representation. In the following sections, we outline our choice of $d(\cdot, \cdot)$ for methods that do (DDPG and TRPO) and do not (DQN) use behavioral policies. In our experiments, we always use $\alpha = 1.01$.

### C.1    A DISTANCE MEASURE FOR DQN

For DQN, the policy is defined implicitly by the $Q$-value function. Unfortunately, this means that a naïve distance measure between $Q$ and $\widetilde{Q}$ has pitfalls. For example, assume that the perturbed policy has only changed the bias of the final layer, thus adding a constant value to each action's $Q$-value. In this case, a naïve distance measure like the norm $\|Q - \widetilde{Q}\|_2$ would be nonzero, although the policies $\pi$ and $\widetilde{\pi}$ (implied by $Q$ and $\widetilde{Q}$, respectively) are exactly equal. This equally applies to the case where DQN as two heads, one for $Q$ and one for $\pi$.

We therefore use a probabilistic formulation[9] for both the non-perturbed and perturbed policies: $\pi, \widetilde{\pi} : \mathcal{S} \times \mathcal{A} \mapsto [0,1]$ by applying the softmax function over predicted $Q$ values: $\pi(s) = \exp Q_i(s) / \sum_i \exp Q_i(s)$, where $Q_i(\cdot)$ denotes the $Q$-value of the $i$-th action. $\widetilde{\pi}$ is defined analogously but uses the perturbed $\widetilde{Q}$ instead (or the perturbed head for $\pi$). Using this probabilistic formulation of the policies, we can now measure the distance in action space:

$$d(\pi, \widetilde{\pi}) = D_{\mathrm{KL}}(\pi \parallel \widetilde{\pi}), \tag{9}$$

where $D_{\mathrm{KL}}(\cdot \parallel \cdot)$ denotes the Kullback-Leibler (KL) divergence. This formulation effectively normalizes the $Q$-values and therefore does not suffer from the problem previously outlined.

We can further relate this distance measure to $\epsilon$-greedy action space noise, which allows us to fairly compare the two approaches and also avoids the need to pick an additional hyperparameter $\delta$. More concretely, the KL divergence between a greedy policy $\pi(s, a) = 1$ for $a = \operatorname{argmax}_{a'} Q(s, a')$ and $\pi(s, a) = 0$ otherwise and an $\epsilon$-greedy policy $\widehat{\pi}(s, a) = 1 - \epsilon + \frac{\epsilon}{|\mathcal{A}|}$ for $a = \operatorname{argmax}_{a'} Q(s, a')$ and $\widehat{\pi}(s, a) = \frac{\epsilon}{|\mathcal{A}|}$ otherwise is $D_{\mathrm{KL}}(\pi \parallel \widehat{\pi}) = -\log\left(1 - \epsilon + \frac{\epsilon}{|\mathcal{A}|}\right)$, where $|\mathcal{A}|$ denotes the number of actions (this follows immediately from the definition of the KL divergence for discrete probability distributions). We can use this distance measure to relate action space noise and parameter space noise to have similar distances, by adaptively scaling $\sigma$ so that it matches the KL divergence between greedy and $\epsilon$-greedy policy, thus setting $\delta := -\log\left(1 - \epsilon + \frac{\epsilon}{|\mathcal{A}|}\right)$.

### C.2   A Distance Measure for DDPG

For DDPG, we relate noise induced by parameter space perturbations to noise induced by additive Gaussian noise. To do so, we use the following distance measure between the non-perturbed and perturbed policy:

$$d(\pi, \widetilde{\pi}) = \sqrt{\frac{1}{N} \sum_{i=1}^{N} \mathbb{E}_s \left[ (\pi(s)_i - \widetilde{\pi}(s)_i)^2 \right]}, \tag{10}$$

where $\mathbb{E}_s[\cdot]$ is estimated from a batch of states from the replay buffer and $N$ denotes the dimension of the action space (i.e. $\mathcal{A} \subset \mathbb{R}^N$). It is easy to show that $d(\pi, \pi + \mathcal{N}(0, \sigma^2 I)) = \sigma$. Setting $\delta := \sigma$ as the adaptive parameter space threshold thus results in effective action space noise that has the same standard deviation as regular Gaussian action space noise.

### C.3   A Distance Measure for TRPO

In order to scale the noise for TRPO, we adapt the sampled noise vectors $\epsilon\sigma$ by computing a natural step $H^{-1}\epsilon\sigma$. We essentially compute a trust region around the noise direction to ensure that the perturbed policy $\widetilde{\pi}$ remains sufficiently close to the non-perturbed version via

$$E_{s \sim \rho_{\widetilde{\theta}}}[D_{\mathrm{KL}}(\pi_{\widetilde{\theta}}(\cdot|s) \| \pi_\theta(\cdot|s))] \le \delta_{\mathrm{KL}}.$$

Concretely, this is computed through the conjugate gradient algorithm, combined with a line search along the noise direction to ensure constraint conformation, as described in Appendix C of Schulman et al. (2015b).

## D   Additional Results on ALE

Figure 7 provide the learning curves for all 21 Atari games.

Table 1 compares the final performance of ES after $1\,000\,\mathrm{M}$ frames to the final performance of DQN with $\epsilon$-greedy exploration and parameter space noise exploration after $40\,\mathrm{M}$ frames. In all cases, the performance is estimated by running 10 episodes with exploration disabled. We use the numbers reported by Salimans et al. (2017) for ES and report the median return across three seeds for DQN.

---

[9]It is important to note that we use this probabilistic formulation only for the sake of defining a well-behaved distance measure. The actual policy used for rollouts is still deterministic.

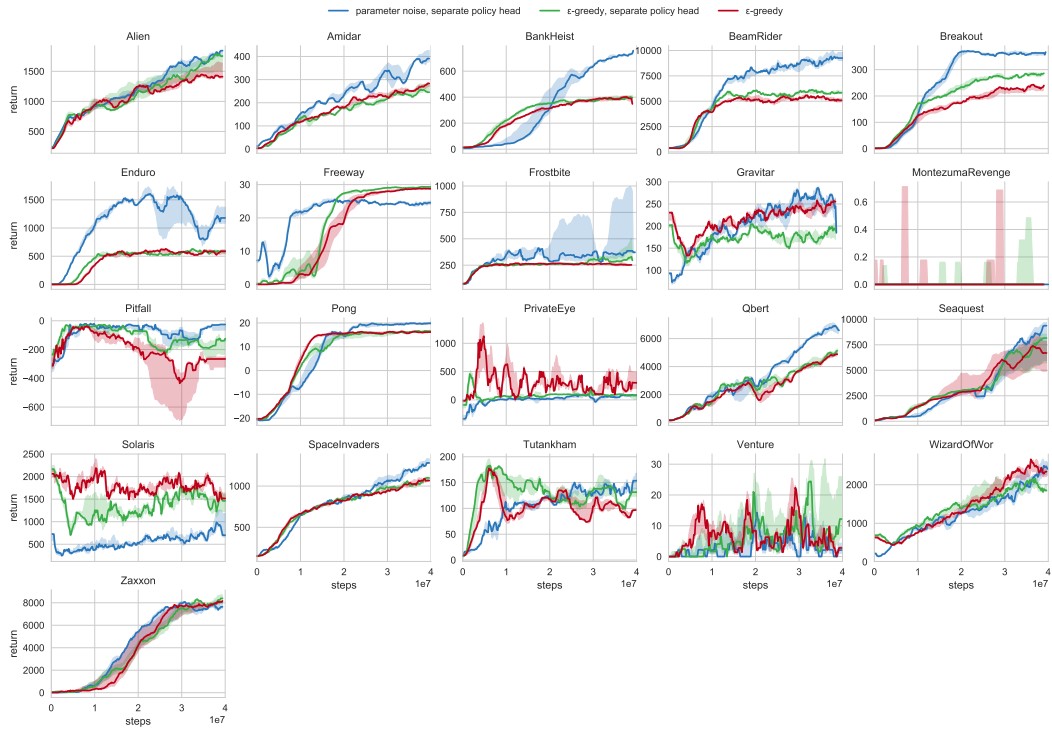

Figure 7: Median DQN returns for all ALE environment plotted over training steps.

Table 1: Performance comparison between Evolution Strategies (ES) as reported by Salimans et al. (2017), DQN with $\epsilon$-greedy, and DQN with parameter space noise (this paper). ES was trained on $1\,000\,\mathrm{M}$, while DQN was trained on only $40\,\mathrm{M}$ frames.

| Game | ES | DQN w/ $\epsilon$-greedy | DQN w/ param noise |
|---|---|---|---|
| Alien | 994.0 | 1535.0 | **2070.0** |
| Amidar | 112.0 | 281.0 | **403.5** |
| BankHeist | 225.0 | 510.0 | **805.0** |
| BeamRider | 744.0 | **8184.0** | 7884.0 |
| Breakout | 9.5 | **406.0** | 390.5 |
| Enduro | 95.0 | 1094 | **1672.5** |
| Freeway | 31.0 | **32.0** | 31.5 |
| Frostbite | 370.0 | 250.0 | **1310.0** |
| Gravitar | **805.0** | 300.0 | 250.0 |
| MontezumaRevenge | **0.0** | **0.0** | **0.0** |
| Pitfall | **0.0** | -73.0 | -100.0 |
| Pong | **21.0** | **21.0** | 20.0 |
| PrivateEye | 100.0 | **133.0** | 100.0 |
| Qbert | 147.5 | **7625.0** | 7525.0 |
| Seaquest | 1390.0 | 8335.0 | **8920.0** |
| Solaris | **2090.0** | 720.0 | 400.0 |
| SpaceInvaders | 678.5 | 1000.0 | **1205.0** |
| Tutankham | 130.3 | 109.5 | **181.0** |
| Venture | **760.0** | 0 | 0 |
| WizardOfWor | **3480.0** | 2350.0 | 1850.0 |
| Zaxxon | 6380.0 | **8100.0** | 8050.0 |

# E  ADDITIONAL RESULTS ON CONTINUOUS CONTROL WITH SHAPED REWARDS

For completeness, we provide the plots for all evaluated environments with dense rewards. The results are depicted in Figure 8.

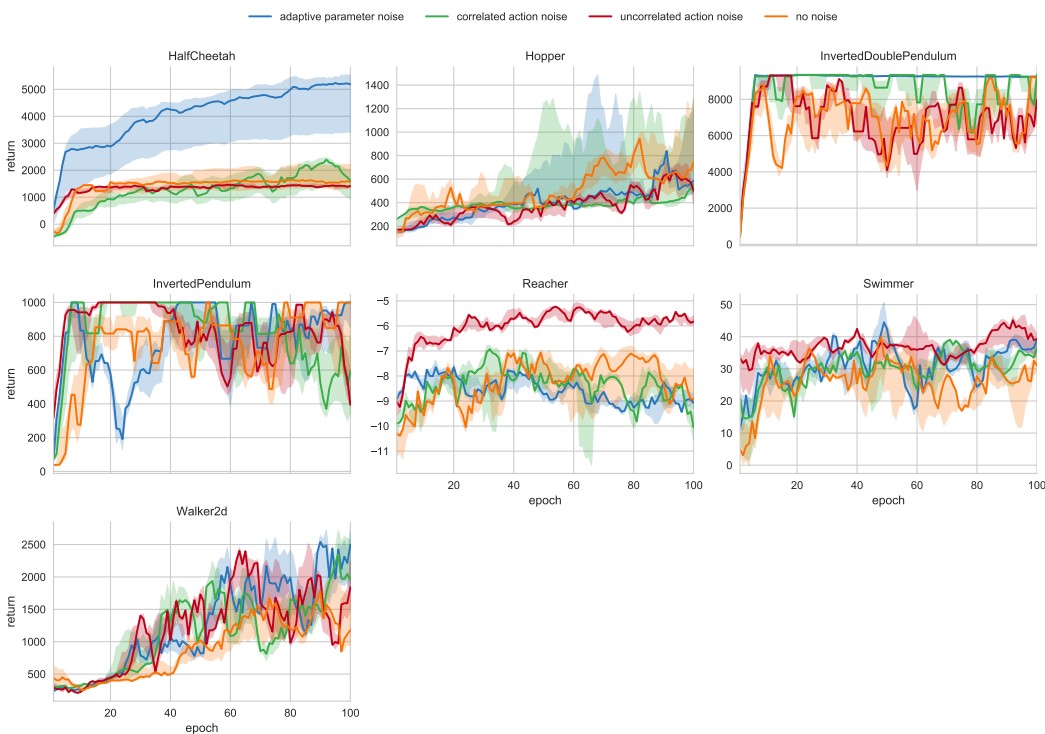

Figure 8: Median DDPG returns for all evaluated environments with dense rewards plotted over epochs.

The results for *InvertedPendulum* and *InvertedDoublePendulum* are very noisy due to the fact that a small change in policy can easily degrade performance significantly, and thus hard to read. Interestingly, adaptive parameter space noise achieves the most stable performance on *Inverted-DoublePendulum*. Overall, performance is comparable to other exploration approaches. Again, no noise in either the action nor the parameter space achieves comparable results, indicating that these environments combined with DDPG are not well-suited to test for exploration.

# F  ADDITIONAL RESULTS ON CONTINUOUS CONTROL WITH SPARSE REWARDS

The performance of TRPO with noise scaled according to the parameter curvature, as defined in Section C.3 is shown in Figure 9. The TRPO baseline uses only action noise by using a policy network that outputs the mean of a Gaussian distribution, while the variance is learned. These results show that adding parameter space noise aids in either learning much more consistently on these challenging sparse environments.

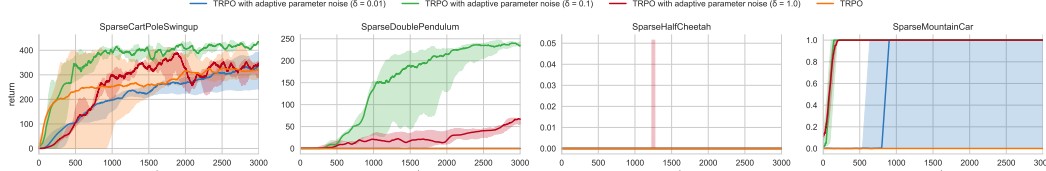

Figure 9: Median TRPO returns with three different environments with sparse rewards plotted over epochs.

