# OpenReview forum: "Parameter Space Noise for Exploration"
_ICLR.cc/2018/Conference — Accept (Poster)_

### Official Review · AnonReviewer3 · 2017-11-09
**How far is parameter noise really going to get us?**

**Rating:** 6
**Confidence:** 4

**Review:**

This paper explores the idea of adding parameter space noise in service of exploration. The paper is very well written and quite clear. It does a good job of contrasting parameter space noise to action space noise and evolutionary strategies.

However, the results are weak. Parameter noise does better in some Atari + Mujoco domains, but shows little difference in most domains. The domains where parameter noise (as well as evolutionary strategies) does really well are Enduro and the Chain environment, in which a policy that repeatedly chooses a particular action will do very well. E-greedy approaches will always struggle to choose the same random action repeatedly. Chain is great as a pathological example to show the shortcomings of e-greedy, but few interesting domains exhibit such patterns. Similarly for the continuous control with sparse rewards environments – if you can construct an environment with sparse enough reward that action-space noise results in zero rewards, then clearly parameter space noise will have a better shot at learning. However, for complex domains with sparse reward (e.g. Montezuma’s Revenge) parameter space noise is just not going to get you very far.

Overall, I think parameter space noise is a worthy technique to have analyzed and this paper does a good job doing just that. However, I don’t expect this technique to make a large splash in the Deep RL community, mainly because simply adding noise to the parameter space doesn’t really gain you much more than policies that are biased towards particular actions. Parameter noise is not a very smart form of exploration, but it should be acknowledged as a valid alternative to action-space noise.

A non-trivial amount of work has been done to find a sensible way of adding noise to parameter space of a deep network and defining the specific distance metrics and thresholds for (dual-headed) DQN, DDPG, and TRPO.

---

> ### Author Response · Authors · 2017-12-15
> **Response to review**
>
> We would like to thank the reviewer for the insightful comments and suggestions.
>
> We do agree that parameter noise alone is not going to solve exploration in reinforcement learning. However, we do feel that it provides an interesting alternative to the still de-facto standard of exploration, which is action space noise like epsilon-greedy or additive Gaussian noise. We think that our paper demonstrates that parameter noise exhibits different behavior that can often result in superior exploration that such simple action space noise exploration methods while being conceptually similarly simple. Furthermore, many recent exploration strategies like intrinsic motivation or count-based approaches augment the reward function with a bonus to encourage exploration but still rely on action space noise for “low-level” exploration. We think that parameter noise could also be an interesting replacement for this low level exploration.
>
> That being said, we do agree that the paper can often seem to overstate the exploration properties of our proposed method. We will carefully revise the manuscript to better present parameter space noise as an interesting alternative to action space noise while emphasizing that it by no means resolves the exploration problem universally.

---

### Official Review · AnonReviewer2 · 2017-11-26
**Nice exploration of parameter noise**

**Rating:** 7
**Confidence:** 4

**Review:**

In recent years there have been many notable successes in deep reinforcement learning. However, in many tasks, particularly sparse reward tasks, exploration remains a difficult problem. For off-policy algorithms it is common to explore by adding noise to the policy action in action space, while on-policy algorithms are often regularized in the action space to encourage exploration. This work introduces a simple, computationally straightforward approach to exploring by perturbing the parameters (similar to exploration in some evolutionary algorithms) of policies parametrized with deep neural nets. This work argues this results in more consistent exploration and compares this approach empirically on a range of continuous and discrete tasks. By using layer norm and adaptive noise, they are able to generate robust parameter noise (it is often difficult to estimate the appropriate variance of parameter noise, as its less clear how this relates to the magnitude of variance in the action space).

This work is well-written and cites previous work appropriately. Exploration is an important topic, as it often appears to be the limiting factor of Deep RL algorithms. The authors provide a significant set of experiments using their method on several different RL algorithms in both continuous and discrete cases, and find it generally improves performance, particularly for sparse rewards.

One empirical baseline that would helpful to have would be a stochastic off-policy algorithm (both off-policy algorithms compared are deterministic), as this may better capture uncertainty about the value of actions (e.g. SVG(0) [3]).

As with any empirical results with RL, it is a challenging problem to construct comparable benchmarks due to minor variations in implementation, environment or hyper-parameters all acting as confounding variables [1]. It would be helpful if the authors are able to make their paper reproducible by releasing the code on publication. As one example, figure 4 of [1] seems to show DDPG performing much better than the DDPG baseline in this work on half-cheetah.

Minor points:
- The definition of a stochastic policy (section 2) is unusual (it is defined as an unnormalized distribution). Usually it would be defined as $\mathcal{S} \rightarrow \mathcal{P}(\mathcal{A})$

- This work extends DQN to learn an explicitly parametrized policy (instead of the greedy policy) in order to useful perturb the parameters of this policy. Instead of using a single greedy target, you could consider use the relationship between the advantage function and an entropy-regularized policy [2] to construct a target.

[1] Henderson, P., Islam, R., Bachman, P., Pineau, J., Precup, D., & Meger, D. (2017). Deep reinforcement learning that matters. arXiv preprint arXiv:1709.06560.

[2] O'Donoghue, B., Munos, R., Kavukcuoglu, K., & Mnih, V. (2016). Combining policy gradient and Q-learning.

[3] Heess, N., Wayne, G., Silver, D., Lillicrap, T., Erez, T., & Tassa, Y. (2015). Learning continuous control policies by stochastic value gradients. In Advances in Neural Information Processing Systems (pp. 2944-2952).

---

> ### Author Response · Authors · 2017-12-15
> **Response to review**
>
> We would like to thank the reviewer for the insightful comments and suggestions.
>
> We agree that reproducibility is an important consideration. The code for DQN and DDPG has already been open-sourced. Unfortunately we cannot directly link to it in the paper due to the double-blind review process. The final version of the paper will include a link to the source code.
>
> We further agree that a stochastic off-policy algorithm such as SVG(0) would be an interesting addition. However, we feel like DQN, DDPG, and TRPO already cover a significant spectrum and we would therefore leave the evaluation with other algorithms like SVG(0) and PPO to future work.
>
> We will also revise the definition of a stochastic policy in the continuous case as suggested by the reviewer.

---

> > ### Comment · AnonReviewer2 · 2018-01-12
> > **read response**
> >
> > I read the author's response and other reviews. I think the author's have addressed most concerns (I'm still curious about the discrepancy in DDPG result). My rating was already positive so I've left it unchanged.

---

### Official Review · AnonReviewer1 · 2017-11-27
**Random exploration at the policy level, rather than the action level - a good paper, but needs to be a bit more careful with the hype.**

**Rating:** 7
**Confidence:** 5

**Review:**

This paper proposes a method for parameter space noise in exploration.
Rather than the "baseline" epsilon-greedy (that sometimes takes a single action at random)... this paper presents an method for perturbations to the policy.
In some domains this can be a much better approach and this is supported by experimentation.

There are several things to like about the paper:
- Efficient exploration is a big problem for deep reinforcement learning (epsilon-greedy or Boltzmann is the de-facto baseline) and there are clearly some examples where this approach does much better.
- The noise-scaling approach is (to my knowledge) novel, good and in my view the most valuable part of the paper.
- This is clearly a very practical and extensible idea... the authors present good results on a whole suite of tasks.
- The paper is clear and well written, it has a narrative and the plots/experiments tend to back this up.
- I like the algorithm, it's pretty simple/clean and there's something obviously *right* about it (in SOME circumstances).

However, there are also a few things to be cautious of... and some of them serious:
- At many points in the paper the claims are quite overstated. Parameter noise on the policy won't necessarily get you efficient exploration... and in some cases it can even be *worse* than epsilon-greedy... if you just read this paper you might think that this was a truly general "statistically efficient" method for exploration (in the style of UCRL or even E^3/Rmax etc).
- For instance, the example in 4.2 only works because the optimal solution is to go "right" in every timestep... if you had the network parameterized in a different way (or the actions left/right were relabelled) then this parameter noise approach would *not* work... By contrast, methods such as UCRL/PSRL and RLSVI https://arxiv.org/abs/1402.0635 *are* able to learn polynomially in this type of environment. I think the claim/motivation for this example in the bootstrapped DQN paper is more along the lines of "deep exploration" and you should be clear that your parameter noise does *not* address this issue.
- That said I think that the example in 4.2 is *great* to include... you just need to be more upfront about how/why it works and  what you are banking on with the parameter-space exploration. Essentially you perform a local exploration rule in parameter space... and sometimes this is great - but you should be careful to distinguish this type of method from other approaches. This must be mentioned in section 4.2 "does parameter space noise explore efficiently" because the answer you seem to imply is "yes" ... when the answer is clearly NOT IN GENERAL... but it can still be good sometimes ;D
- The demarcation of "RL" and "evolutionary strategies" suggests a pretty poor understanding of the literature and associated concepts. I can't really support the conclusion "RL with parameter noise exploration learns more efficiently than both RL and evolutionary strategies individually". This sort of sentence is clearly wrong and for many separate reasons:
    - Parameter noise exploration is not a separate/new thing from RL... it's even been around for ages! It feels like you are talking about DQN/A3C/(whatever algorithm got good scores in Atari last year) as "RL" and that's just really not a good way to think about it.
    - Parameter noise exploration can be *extremely* bad relative to efficient exploration methods (see section 2.4.3 https://searchworks.stanford.edu/view/11891201)


Overall, I like the paper, I like the algorithm and I think it is a valuable contribution.
I think the value in this paper comes from a practical/simple way to do policy randomization in deep RL.
In some (maybe even many of the ones you actually care about) settings this can be a really great approach, especially when compared to epsilon-greedy.

However, I hope that you address some of the concerns I have raised in this review.
You shouldn't claim such a universal revolution to exploration / RL / evolution because I don't think that it's correct.
Further, I don't think that clarifying that this method is *not* universal/general really hurts the paper... you could just add a section in 4.2 pointing out that the "chain" example wouldn't work if you needed to do different actions at each timestep (this algorithm does *not* perform "deep exploration").

I vote accept.

---

> ### Author Response · Authors · 2017-12-15
> **Response to review**
>
> We would like to thank the reviewer for the insightful comments and suggestions.
>
> We will update section 4.2 to better reflect the limitations of our proposed method and to clarify that parameter noise is by no means a universally applicable strategy with guarantees. In particular, we will include a paragraph in the chain environment discussion to highlight that parameter noise works well here due to the simplicity of the optimal strategy. We will further clarify that this experiment was intended to highlight the difference in behavior between epsilon-greedy exploration and parameter noise and that it is clearly a toy problem that should not be interpreted as a claim that parameter noise exploration results in universally better exploration.
>
> We will also revise the text that is concerned with the discussion of ES and RL. We do agree that parameter noise in general is by no means a novel concept and will revise accordingly. We will further clarify that the scope of the proposed approach is to make parameter noise work in the context of deep reinforcement learning and that our comparison is meant to highlight the advances in sample complexity compared to the method proposed by Salimans et al. (2017).
>
> Generally speaking, we will revise our paper to better reflect what parameter noise really is: A conceptually simple replacement for simple exploration strategies like epsilon-greedy that often results in better exploration than these baselines. However, by no means is parameter noise a universally applicable method with guarantees like RLSVI or E^3. We will add language to clearly state this.

---

### Author Response · Authors · 2018-01-03
**Revised manuscript**

Dear reviewers, a revised manuscript that takes your feedback into account has been uploaded.

---

### Public Comment · (anonymous) · 2018-01-06
**Summary of a reproducibility study**

We performed a brief reproducibility study for this paper. The objective was to examine the potential benefits of parameter noise and measure its robustness to changes in hyperparameters and network configuration.

We found that the implementation we used (which is linked in the paper below) currently has a bug present in the master branch which disrupts the adaptive scaling algorithm for Q-learning, causing explosion of the noise variance in some environments (namely, Zaxxon). This is fixed by pull request #143 of the implementation repository and was later noted as issue #157, both of which are open as of the time of writing. Our results were generated by a patched version which merges pull request #143 with the master branch.

To summarize the report:

(1) For the initial regime of the continuous control environments, we observed the behaviour mentioned in this paper where action noise policies will only flip the HalfCheetah agent on its back and slowly work their way forward, while the parameter noise policy will eventually learn a more realistic and better performing gait. Owing to time constraints, these experiments were left as incomplete; we are unable to present a quantitative evaluation of parameter noise with respect to continuous control.

(2) Two of the improvements which had been inferred to be orthogonal to parameter space noise (dueling networks and prioritized replay), at least in certain ALE environments, appear to improve the performance of epsilon-greedy policies without improving the performance of policies which use parameter noise. So, on certain environments, adding parameter space noise may not improve an epsilon-greedy policy that already uses dueling networks and prioritized replay. While our results on ALE were limited by time and computational constraints, the implementation authors provide their own results (which are linked in section 4.1 of our report) that corroborate this point more strongly. This effect is especially visible in the Enduro environment where the unimproved parameter noise policy dominated the unimproved epsilon-greedy policy by a considerable margin (as displayed in Figure 1 of the paper under examination) but the improved epsilon-greedy policy performed either better or near-identically to both the improved and unimproved parameter noise policies (as displayed in both the implementation authors' and our results).

(3) From the ablation studies on Walker2D, we saw that removing layer normalization may degrade performance of the parameter noise policy. We also observed that varying the noise scale parameter by any more than an order of magnitude in either direction causes loss of the ability to learn.

The full report is available here:
https://github.com/c-connors/param-noise-repr/blob/master/parameter-space-noise.pdf

---

> ### Author Response · Authors · 2018-01-08
> **Thank you for conducting this study**
>
> I would like to thank you for conducting this study and evaluating the behavior of parameter space noise with duelling networks and prioritized replay. It is interesting that both seem to not help in this case.
>
> I would also like to note that the bug that you mention was only introduced while we refactored our code to be releasable (this was necessary due to the original code having a heavy dependance on our internal infrastructure). The experiments presented in our paper were not affected by it and the scaling was handled correctly.

---

### Decision · Program_Chairs · 2018-01-29
**ICLR 2018 Conference Acceptance Decision**

**Decision:**

Accept (Poster)

**Comment:**

This paper proposes adding noise to the parameters of a deep network when taking actions in deep reinforcement learning to encourage exploration.  The method is simple but the authors demonstrate its effectiveness through thorough empirical analysis across a variety of reinforcement learning tasks (i.e. DQN, DDPG, and TRPO).  Overall the paper is clear, well written and the reviewers enjoyed it.  However, a common trend among the reviews was that the authors overstated their claims and contributions.  The reviewers called out some statements in particular (e.g. the discussion of ES and RL) which the authors appear to have addressed when comparing their revisions (thank you).  Overall, a clear, well written paper conveying a simple but effective idea for exploration that often works across a variety of RL tasks.  The authors also released open-source code along with their paper for reproducibility (as evidenced by the reproducibility study below), which is appreciated.

Pros:
- Clear and well written
- Thorough experiments across deep RL domains
- A simple strategy for exploration that is effective empirically

Cons:
- Not a panacea for exploration (although nothing really is)
- Claims are somewhat overstated
- Lacks a strong justification for the method other than that it is empirically effective and intuitive